# Isolation and Characterization of *Erianthus arundinaceus* Phosphate Transporter 1 (PHT1) Gene Promoter and 5′ Deletion Analysis of Transcriptional Regulation Regions under Phosphate Stress in Transgenic Tobacco

**DOI:** 10.3390/plants12213760

**Published:** 2023-11-03

**Authors:** Murugan Naveenarani, Huskur Kumaraswamy Mahadeva Swamy, Sakthivel Surya Krishna, Channappa Mahadevaiah, Ramanathan Valarmathi, Markandan Manickavasagam, Muthukrishnan Arun, Govindakurup Hemaprabha, Chinnaswamy Appunu

**Affiliations:** 1Division of Crop Improvement, Indian Council of Agricultural Research-Sugarcane Breeding Institute, Coimbatore 641007, Tamil Nadu, India; naveenamurugan03@gmail.com (M.N.); maddygkvk@gmail.com (H.K.M.S.); surikrish140@gmail.com (S.S.K.); msheshadri@gmail.com (C.M.); r.valarmathi@icar.gov.in (R.V.); g.hemaprabha@icar.gov.in (G.H.); 2Bharathidasan University, Tiruchirappalli 620024, Tamil Nadu, India; 3Division of Vegetable Crops, Indian Institute of Horticultural Research, Bengaluru 560089, Karnataka, India; 4Department of Biotechnology, Bharathidasan University, Tiruchirappalli 620024, Tamil Nadu, India; manickbiotech@gmail.com; 5Department of Biotechnology, Bharathiar University, Coimbatore 641046, Tamil Nadu, India; arun@buc.edu.in

**Keywords:** *Erianthus arundinaceus*, high-affinity phosphate transporter, PHT1;2, promoter analysis, tobacco transgenic, Pi stress

## Abstract

Phosphorus deficiency highly interferes with plant growth and development. Plants respond to persistent P deficiency by coordinating the expression of genes involved in the alleviation of stress. Promoters of phosphate transporter genes are a great choice for the development of genetically modified plants with enhanced phosphate uptake abilities, which improve crop yields in phosphate-deficient soils. In our previous study, the sugarcane phosphate transporter PHT1;2 gene showed a significantly high expression under salinity stress. In this study, the *Erianthus arundinaceus EaPHT1;2* gene was isolated and characterized using various in silico tools. The deduced 542 amino acid residues have 10 transmembrane domains, with a molecular weight and isoelectric point of 58.9 kDa and 9.80, respectively. They displayed 71–96% similarity with *Arabidopsis thaliana*, *Zea mays*, and the *Saccharum* hybrid. To elucidate the function of the 5′ regulatory region, the 1.1 kb promoter was isolated and validated in tobacco transgenics under Pi stress. The EaPHT1;2 promoter activity was detected using a β-glucuronidase (GUS) assay. The EaPHT1;2 promoter showed 3- to 4.2-fold higher expression than the most widely used CaMV35S promoter. The 5′ deletion analysis with and without 5′ UTRs revealed a small-sized 374 bp fragment with the highest promoter activity among 5′ truncated fragments, which was 2.7 and 4.2 times higher than the well-used CaMV35S promoter under normal and Pi deprivation conditions, respectively. The strong and short promoter of EaPHT1;2 with 374 bp showed significant expression in low-Pi-stress conditions and it could be a valuable source for the development of stress-tolerant transgenic crops.

## 1. Introduction

Phosphate (P) is a crucial nutrient essential for plant growth, physiological responses, and stress tolerance, and its efficient uptake is vital for crop productivity. Inorganic phosphate (Pi) is the main source of P for plants; however, it can be challenging for them to absorb because it is in complex with metal ions in the soil [1]. Pi is transported in plants through root cells, xylem, and shoot tissues, involving specific Pi transporters. Phosphate transporters help maintain phosphate homeostasis within plants. These transporters are responsible for Pi uptake, translocation, and remobilization [1,2]. The availability of P in the soil is often limited and therefore plants have evolved various mechanisms to acquire and transport phosphate ions (Pi) from the soil to meet their physiological needs [1,3]. Under limited Pi availability, plants employ various strategies to adapt to their environment. These adaptations encompass symbiotic approaches, alterations in root architecture, the release of organic acids, and the production of acid phosphatases on the part of the roots [4,5]. However, under stress conditions, plants exhibit adaptive mechanisms to optimize nutrients, but they are not robust enough to prevent the struggle against detrimental effects like stress signaling and stress response mechanisms [6,7,8,9]. Soil conditions, climate change, fertilizer costs, and plant use efficiency are critical factors that influence P availability to plants [10,11]. Nutrient deficiency, coupled with abiotic stresses such as drought, salinity, and temperature, can also have profound effects on plants, impacting their growth and development and reducing crop yields, which in turn affects the global food security [12].

The phosphate transporter 1 (PHT1) family has been the most extensively studied of the plant phosphate transporters. PHT1 is a plasma membrane protein that plays the main role in direct Pi uptake from the soil by plant roots. The movement of Pi via PHT1 is driven by plasma membrane H^+^-ATPase, which belongs to the family of H^+^/Pi symporters within the major facilitator superfamily (MFS) and possesses high Pi affinity and strong expression in roots, being especially up-regulated in Pi-deprived plants [13,14]. Since 1996, functional characterization of PHT1 members has been conducted using mutant lines and transgenic plants. Lately, the precise functions of these members have been examined using various single and multiple mutants produced via synthetic microRNA silencing. Recent studies using GWAS and in silico analyses of plant genomes revealed PHT1 genes in several crop plants like *Hordeum vulgare* (barley), *Oryza sativa* (rice), *Zea mays* (maize), *Setaria italica* (foxtail millet), *Solanum lycopersicum* (tomato), *Populus trichocarpa* (poplar), *Malus domestica* (apple), *Triticum aestivum* (wheat), *Sorghum bicolor* (L.) Moench (sorghum), *Camellia oleifera Abel*. (tea tree oil), *Camelina sativa* (false flax), and *Medicago truncatula* (barrel medic) [15,16,17,18,19,20,21,22,23,24,25,26], while their precise functions are still under investigation.

In plants, the majority of the PHT1 genes are transcriptionally activated by Pi deficiency and up-regulated in the roots, shoots, or both. These responses in the PHT genes are regulated by the transcription factors (TFs), linked to consensus cis-acting sequences. The well-known phosphate starvation response (PHR) transcription factors, from the MYB-CC family, are the key regulators of Pi starvation signaling. The PHR TFs regulate phosphate-starvation-induced (PSI) gene expression by binding to P-responsive P1BS elements (GNATATNC) in their promoter regions [27,28,29,30,31]. Other regulatory components like microRNA miR339, PHF1, SIZ1, PHR1, MYB62, and WRKY75 have also been reported in *Arabidopsis* and rice [32,33]. Beyond Pi starvation signaling, PHT genes are also regulated transcriptionally, indicating a more complex regulatory mechanism than anticipated. Several cis elements have been reported for their role in the transcriptional signaling of the expression of Pi-starvation-responsive genes. Cis-acting elements like P1BS, MBS, W-box, G(E)-box, NIT 2, and PHO elements were reported to be associated with the responsiveness of PHT1s and PSI genes [1,34,35,36,37]. 

Sugarcane holds significant economic importance as it is cultivated in approximately 121 nations, and produces an impressive cane yield of 70.9 tons per hectare [38]. This crop plays a pivotal role in food and industrial uses, as nearly 80% of the sugar consumed globally is made from sugarcane. Sugarcane productivity is significantly limited by phosphorus (P), which is crucial for vegetative development and crop durability. P fertilizers address P deficiency in high-input agricultural systems, ~120 kg P ha^−1^ annually; this legacy phosphorus can indeed become a valuable resource if managed effectively within crop systems [12,39]. Brazil, the largest sugarcane producer, has seen significant growth in stalk yields since 1975 due to improved soil quality, plant breeding, and crop agronomy. However, Brazil’s sugarcane production faces challenges due to low phosphorus availability, affecting 20% of fertilizer consumption and making nutrient security vulnerable to future scarcity [40]. 

We functionally characterized the high-affinity phosphate transporter 1 gene promoter from sugarcane (*Saccharum* spp. hybrid) based on our previous report [41,42]. In this current study, we isolated the full-length coding sequence and promoter sequences of the *EaPHT1;2* gene from *Erianthus arundinaceus*, a wild genus of the *Saccharum* spp. The *EaPHT1;2* gene and promoter region were isolated and characterized using various bioinformatics tools and further, the EaPHT1;2 promoter was functionally validated in tobacco. The full-length EaPHT1;2 promoter and a series of truncated promoters with and without 5’ untranslated regions (UTRs) were fused to the β-glucuronidase (GUS) reporter gene to identify their expression under Pi stress. We found that in transgenic tobacco, the shortest promoter of 374 bp has significantly strong GUS expression under low-Pi-stress conditions. 

## 2. Results

### 2.1. Isolation and Sequence Analysis of EaPHT1;2 Gene from E. arundinaceus

In our previous study, the PHT1;2 gene sequence was obtained from the sugarcane genome of *Saccharum* spp. hybrid (Sh) cultivar R570; thus, a gene-specific primer pair designed based on the conserved regions in *Saccharum* spp. and *Z. mays* was used for isolation of the PHT1;2 gene from the *E. arundinaceus* genomic DNA (Figure 1A). The amplicon was cloned and sequenced for further characterization (Figure 1B). BLAST analysis of EaPHT1;2 showed 71–96% similarity with ZmPHT1, SbPHT1, ShPHT1, and AtPHT1. A phylogenetic analysis using MEGA was carried out to confirm the evolutionary relationships among these PHT1 proteins (Figure 1C). The amplified full-length coding sequence of EaPHT1;2 resulted in an open reading frame (ORF) of 1626 bp, which encodes a protein of 542 amino acid residues with a molecular weight of 58.9  kDa (Appendix A). Further physicochemical properties were also predicated using the ProtParam tool (Appendix A and Appendix A). The results showed that the EaPHT1;2 protein has a theoretical isoelectric point (pI) of 9.80 and a grand average of hydropathicity (GRAVY) of 0.019. Analysis of active domains in the coding sequence confirmed the presence of a functional domain with signature sequences characteristic of the major facilitator superfamily (MFS, PS50850) (Figure 2A). The protein was anticipated to contain 10 transmembrane domains (TMs), with a hydrophilic loop in the central part of the domains initially identified using the TMHMM server v. 2.0 and further verified using a Kyte–Doolittle hydropathy plot (Figure 2B,C). The post-translation modification analysis revealed a total of 39 phosphorylation sites with a distribution of 24:14:1 Ser:Thr:Tyr sites, and most of the discovered phosphopeptides were monophosphorylated. N-glycosylation site analysis identified a potential NSTT site for N-linked glycosylation at the 425 amino acid position (Appendix A). 

### 2.2. Structural Prediction of the EaPHT1;2 Protein 

The secondary structure prediction using SOPMA (Appendix A) revealed a predominance of an alpha helix of 37.45% followed by random coil (35.79%), extended strand (17.71%), and beta turn (9.04%). As presented in Figure 3A, the 2D structure resulting from the PsiPred server exhibited 22 α-helices, 1 β-strands, and 20 coil structures. Analysis of the PHT1-protein-specific signature sequence (GGDYPLSATIxSE) of EaPHT1;2 in comparison with the PHT1 proteins of *Z. mays*, *S. bicolor*, *Arabidopsis*, and the *Saccharum* hybrid revealed the existence of a conserved identical GGDYPLSATIMSE sequence in the protein (Figure 3B). The homology modeling of the protein was performed using Phyre2 and SWISS-MODEL. The 3D model resulting from Phyre2 analysis, with c4j05A as the template and having a confidence value of 100%, indicated 77% coverage (420 residues) of the EaPHT1;2 sequence. However, the protein ID for SWISS-MODEL was 7sp5.1.A and showed a 22-523 sequence range and a Q-mean of 0.6, indicating that the developed homology model of EaPHT1;2 had 92.4% coverage (Figure 4A,B). The resulting model was validated using Ramachandran plot analysis, and 86% and 89% of the residues were located in the most favorable area for the Phyre2 and SWISS-MODEL structures, respectively (Figure 4C,D). As a result, the model produced using SWISS-MODEL is more acceptable than that produced using Phyre2 (Figure 4E,F). Qualitative evaluation of the EaPHT1;2 protein 3D structure was also carried out using the ProSA web server (Appendix A). The structure of the EaPHT1;2, in a complex with a substrate of inorganic phosphate ions (PO4), was determined to have a 4 Å resolution, with six phosphate binding sites (A:Y.152, A:Q.179, A:Q.179, A:W.322, A:W.322, A:Y.330) interacting via hydrogen bonds (2.44–4.0 Å distances) with the phosphate in chain A (Figure 4G,H). The electrostatic surface analysis of the EaPHT1;2 protein was visualized using UCSF Chimera.

### 2.3. Protein Association Network and Subcellular Localization of Analysis

The EaPHT1;2 protein was subjected to a protein–protein interaction (PPI) study using STRING, which was predicated based on interactions between homologous proteins from *S. bicolor*, *Z. mays*, and *Arabidopsis*. The findings showed that the EaPHT1;2 protein interacts with more than 10 possible protein partners of three different genomes. The PPI interaction results indicated that for the EaPHT1;2 (red-colored node—Pht1, Sb01g020570.1, and PHT1;1 in Appendix A) protein, the majority of these proteins belong to the phosphate transporter family, like the MFS transporter, phosphodiesterase precursor, SPX domain protein, sodium-phosphate symporter, solute carrier, and sugar transporter. Between the ZmPHT1 and EaPHT1;2 interactions, 20 proteins were predicted as functional partners, two of the members (umc1363a and expB8) belonging to the expansin family, and other proteins like cps2, NAR2.2, and pco091084 belonging to the chaperonin (HSP60) family, the nitrate transporter family, and histidine acid phosphatase family of proteins, respectively. The PPI analysis revealed that the EaPHT1;2 protein interacted with the phosphate transporter protein families (Appendix A). The PPI interaction of EaPHT1 with the PHT1 protein in *Arabidopsis*, a member of the MYB-CC family, acts as a major integrator of phosphate starvation responses by regulating FER1 expression in response to phosphate deficiency, thereby connecting iron and phosphate homeostasis. RNS1, a member of the ribonuclease T2 family, responds to inorganic phosphate starvation and wound-induced signaling. SPX1 and PHR1 are both in different ways involved in phosphate starvation in plants (Appendix A). With respect to sorghum-based EaPHT1 interactions, two members, namely actin 7 and actin7-like proteins, were actively engaged in numerous sorts of cell motility processes and are universally found in all eukaryotic cells (Appendix A). The subcellular localization analysis of the EaPHT1;2 protein was predicted using DeepLoc1.0. The EaPHT1;2 protein localization of various organelles was distinguished using approximate values, as are presented in Appendix A. The analysis anticipated that the EaPHT1;2 protein was more likely to be located in the cell membrane (Appendix A). The QuickGO analysis also revealed that the EaPHT1;2 protein is a cellular component present in the cell peripheral region, particularly in the plasma membrane (Appendix A).

### 2.4. Isolation of PHT1 Promoter Region Using a Genome Walking Method

The 5′ promoter region present upstream of the translation initiation codon (ATG) of the *EaPHT1;2* gene was isolated using the random amplification of genome ends (RAGE) approach. Five different genomic DNA libraries were generated by digesting the Erianthus genomic DNA using blunt-end restriction enzymes (Figure 5A) and ligated with the adapter sequence used as the template for the primary PCR reaction. The primary PCR was performed using the ASP I and GSP I primers (primers are listed in Appendix A). Further, the secondary PCR reactions were carried out using the primary PCR amplified product as a template (Figure 5B) using the ASP II and GSP II primers. The amplicon of a putative promoter region ~1500 bp, obtained from a genomic DNA library using DraI, was isolated, cloned, and sequenced. Finally, after sequencing, 1102 bp of the 5′ upstream region of the EaPHT1;2 promoter region was obtained via sequence alignment with the gene sequences. 

### 2.5. In Silico Analysis of Promoter 

The EaPHT1;2 promoter sequence was analyzed by means of online tools and software. To identify the transcription start site, the promoter sequence of EaPHT1;2 was analyzed by using NNPP. The closest predicted TSS on the promoter sequence begins 83 base pairs ahead of the ATG codon (Figure 5C). The analysis based on the PLACE and PlantCARE tools indicated the existence of at least 28 different types of regulatory motifs, including conserved motifs such as the TATA box, CAAT box, and Pi-starvation-responsive motif P1BS (GNATATNC). Root-specific motifs like ASF1, OSE1, OSE2, RAV1AAT, SURE core, TAPOX1, and ARFAT were also mapped in Figure 5D. The EaPHT1;2 promoter region was also compared with the ZmPHT1 and ShPHT1;2 promoter regions (Appendix A). In *Z. mays* and *E. arundinaceus*, two and one conserved regions were spotted with different TFs, respectively (Appendix A). Multiple cis elements that enable a stress response; inducible or tissue-specific expressions like ABA, salinity, and dehydration; light-responsive and hormone-responsive elements; and pathogen/elicitor-sensitive motifs were found at numerous positions in the EaPHT1;2 promoter sequence. 

### 2.6. Expression Analysis of EaPHT1;2 Promoters in Tobacco Transgenic Events 

To test the promoter activity, a total of six constructs, namely a full-length promoter (pFL) and deletions pD1–pD5, were generated and transformed in tobacco. Putative transgenic plants were validated and confirmed using two sets of specific primer pairs. The putative T0 generation tobacco transgenic events were subjected to a GUS histochemical assay and the untransformed wild-type plant served as a control. GUS expression analysis of the tobacco events showed a strong constitutive expression in all tissues (Figure 6A). The histochemical assay findings revealed that pFL::GUS was efficient in driving the expression of genes in all tissues. Comparatively, the pFL::GUS promoter had a higher level of GUS expression than pCaMV35S::GUS, and in contrast, it is challenging to detect a substantial variation in expression intensities by using only GUS histochemical analysis. Therefore, fluorometric GUS assays of the leaf, stem, and root tissues of 60-day-old plants (GUS-positive events) were taken to elucidate the difference in the GUS expression levels between the pFL::GUS and pCaMV35S::GUS. Enhanced GUS activities was recorded in the roots, stems, and leaves of pFL::GUS compared to pCaMV35S::GUS promoter transgenic plants. MUG assay revealed GUS activity 1.5 times higher in pFL::GUS events compared to those with the CaMV35S promoter (Figure 6C). Therefore, to determine the activities of the various regulatory regions of the pFL, a series of 5′-deletion analysis tests with and without 5′ UTR regions were carried out (Figure 6B). The study showed that all EaPHT1;2 promoter fragments drove equally regulated gene expression in all the tissues, despite variation in the GUS staining intensity. The fluorometric GUS activity showed 1.2 and 2.6 times increased expression in deletion promoters pD1 and pD2, and the promoter’s pD3, pD4, and pD5 showed similar or slight expression as compared to that with the CaMV35S promoter, respectively. The result revealed that plants with the 5′ UTR region in the deletion constructs pD1 and pD2 showed a substantial increase in GUS expression compared to the pD3 to pD5 promoters without the 5′ UTR region. This confirmed that 5′ deletion analysis with or without the 5′ UTR region did not result in function loss.

### 2.7. EaPHT1;2 Promoter Activity in Pi Stress

Seedlings of the T1 generation 20 days old were subjected to low (0.1 mM) and high (1 mM) Pi stress. The quantification of the GUS activity in both treated and untreated transformed plants was performed. With an increase in Pi concentration (high Pi stress), the activities of all the promoters showed decreased GUS expression; however, the GUS activity of transgenic plants under low Pi stress displayed significantly increased GUS activity, compared to the untreated control transgenic seedlings. The results showed that under low Pi conditions, the pD2 promoter had an expression 4.2 times higher when compared to the CaMV35S promoter, and an expression 1.4 times higher than the untreated transgenic control (Figure 6C). In the CaMV35S promoter, no significant expression was detected in both high and low Pi conditions. Notably, the pD2 sequence still displayed the highest promoter activities among other promoters under both high and low Pi conditions. 

### 2.8. Relative GUS Expression Analysis of the EaPHT1;2 Promoter Using RT-PCR 

For qRT-PCR analysis, T1 transgenic events with higher levels of GUS expression were used. The results indicate that the pFL had a 2.5-fold higher expression than the CaMV35S promoter under normal conditions. The GUS expression level increased with low Pi treatment and thus, the findings provided evidence that the EaPHT1;2 gene promoter has a strong expression under low-Pi-stress conditions. Therefore, the deletion analysis of the promoter might aid a deeper understanding of the tissue-specific stress or minimal promoter region analysis. The expression patterns of five EaPHT1;2 promoters under normal, high, and low Pi stress were subsequently investigated using qRT-PCR in order to validate the stress-induced expression level of the pFL and five pD1 to pD5 deletion promoters in comparison with the pCaMV35s promoter. According to the findings, the pFL and pD2 genes had higher levels of expression than the pCaMV35s promoter, and other EaPHT1;2 promoters (Figure 6D). The pFL showed enhanced expression under low phosphate: the expression levels were 2.5- and 3.0-fold higher than that of the pCaMV35s promoter under normal and Pi stress. The expression of pD1 was up-regulated 4.2-fold under low Pi stress, and 2.7-fold under normal conditions. In contrast, the promoters without 5′ UTR regions, the pD3, pD4, and pD5 promoters, showed 1.0- to 1.5-fold increased expression under low Pi stress (Figure 6D). Based on these findings, we hypothesized that the phosphate transporter gene promoter had a strong gene expression in low-Pi conditions, and in addition, this study showed the importance of the 5′ UTR region, which plays a crucial role in promoter regulatory components and acts as the positive enhancer that regulates the expression of the promoter region in response to diverse abiotic stresses.

## 3. Discussion

Phosphorus (P) is essential for plant growth, development, and abiotic stress tolerance, and its deficiency can limit crop productivity and affect yield and quality. Plants use various mechanisms to cope with low phosphorus availability by optimizing phosphorus uptake and usage. To satisfy this purpose, phosphate transporter 1 (PHT1) genes play a crucial role in phosphorus uptake by transporting inorganic phosphate from the soil, which is extensively studied in several plant species. As far as we know, the functional characterization of the PHT1 gene promoter in sugarcane has not been studied to date. The first study on a genome-wide analysis of the PHT gene family in sugarcane was reported by us in 2022 [41], revealing that the gene PHT1;2 has a higher level of expression under salinity stress conditions in sugarcane roots. 

In this study, we validated the EaPHT1;2 gene promoter under Pi stress. The *EaPHT1;2* gene was isolated and characterized using various bioinformatic tools. The *EaPHT1;2* gene belongs to the high-affinity phosphate transporter family and shares a significant sequence similarity with other plant PHT1 family transporters, particularly ZmPHT1 (*Z. mays*), followed by the SbPHT1 (*S. bicolor*) and AtPHT1 (*A. thaliana*) genes [43]. Sugarcane EaPHT1;2 had the most average length and weight of any of the plant phosphate transporters, at 542 aa and 58.9 kDa. The isoelectric point of EaPHT1;2, is similar to most PHT1 members, and it is conceivable that they are mostly present in plasma membranes. Based on the results obtained and earlier reports, the majority of PHT1s appear to be membrane-bound, which is essential for ion transport [44]. The multiple sequence alignment of the EaPHT1;2 protein with the reference genomes was carried out by utilizing the Clustal X program, and as illustrated in Figure 3B, the EaPHT1;2 transporter is identical to other PHT1 proteins, which possess the characteristic motif sequence GGDYPLSATIMSE (GGDYPLSATIxSE) [45,46,47]. The secondary structure showed 10 transmembrane domains with a hydrophilic loop in the central part of the domains, 39 phosphorylation sites, and 1 N-glycosylation site [13,24,48,49]. The 3D structure of many plant PHT proteins is still not available for comparative analysis with the phosphate transporter genes; therefore, according to the homology in SWISS-MODEL, a 7sp5.1.A template was chosen to construct the 3D structure with 92.4% sequence coverage, followed by validation, and the binding sites of the EaPHT1;2 protein structure were analyzed. Figure 3B illustrates the structure of the EaPHT1;2 protein, which has a 4 Å resolution with six phosphate binding sites facing inward to the protein. Overall, the amino acid sequence and 3D structure of the EaPHT1;2 protein exhibit a general conservation with PnPht1;2, CmPht1;2, and the other reported PHT1 proteins [47,49,50].

A 1102 bp of the 5′ regulatory region was isolated from EaPHT1;2. The cis-regulatory elements present in the promoter region govern several essential roles, including biotic and abiotic stress, hormonal stress, developmental process regulation, etc., [51,52]. The in silico analysis of the 5′ regulatory region found in the gene promoters aided an understanding of the molecular process behind the gene expression by using PLACE and PlantCARE. The examination of the EaPHT1;2 promoter region revealed a wide range of previously recognized elements. The conserved promoter elements like the P1BS element are mostly present in all of the Pht1 promoters in one or multiple copies. And in the EaPHT1;2 promoter, two copies of the P1BS element were discovered, which is similar to in TaPH2 and HvPht1;2 [53,54]. The P1BS element is a conserved Pi-starvation-responsive element, typically found as a conserved signaling route for the Pi starvation response in plants. The P1BS element acts as the binding site for the PHR1 protein, a MYB transcription factor, presumably controlling the Pi starvation response in plants [55]. The substantial conservation of this candidate motif in other Pi transporter promoters in Arabidopsis, rice, barley, wheat, eggplant, and soy bean indicates that it is functionally conserved and may play a role in boosting root expression under Pi-deficient circumstances [53,55,56,57,58,59,60]. Secondly, a number of root-specific regulatory elements, including ROOTMOTIFTAPOX1, OSE1ROOTNODULE, and OSE2ROOTNODULE, and other tissue-specific elements like CACTFTPPCA1, CAATBOX1, GTGANTG10, CCGTCC-box, and CAT-box were also discovered. Another major cis element investigated intensively in the PHT gene promoters, the most conserved element next to P1BS, is W-Box, which positively regulates Pi starvation and might be present in one or more copies in their promoter regions. Within the EgPHT1 promoter, many W-box sequences were found. Similarly, the EaPHT1;2 promoter also have 10 W-Box sequences [60]. Along with the conserved cis element motifs, the EaPHT1;2 gene promoter contains various stress-responsive elements such as LTRE, MYB, MYC, AuxR, and GATA sites.

A previous study on wheat described that in response to Pi starvation, the TaPHT1;2 gene was primarily expressed in the roots of wheat seedlings, and its expression levels were higher in the roots of P-deficiency-tolerant wheat varieties [61]. In Arabidopsis, the PHT1 family were expressed at the root–soil interface, which is involved in phosphate absorption and translocation in plants [26]. In Arabidopsis, the Pht1;2 phosphate transporter regulates root-specific gene expression, which is significantly enhanced by Pi shortage [62]. High-level transgenic expression may be delivered to almost all tissues and in all developmental stages in plants via powerful constitutive promoters. Currently, CaMV35s, rice actin (OsAct), and maize ubiquitin (Mubi) are the three mostly widely employed promoters in plant genetic transformation [63,64]. In addition, stress-inducible promoters and tissue- or organ-specific promoter are now widely studied for their desired expression in plants. Associated with abiotic stress, nutrient stress also has a huge impact on plant growth and development, but the combined abiotic stress and crop nutrient stress still remains unrevealed [12]. 

In this study, the EaPHT1;2 promoter was functionally characterized under Pi stress. Previous research discovered that high expression of the Pht1;2 promoter is preferentially induced in the epidermis of Pi-deficient roots [62]. The full-length promoter region of the EaPHT1;2 promoter showed highly constitute expression, which is a relatively 2.5-fold higher expression than for the CaMV35S promoter in transgenic tobacco plants under normal conditions, and possessed a higher expression under low Pi stress. pFL and pD2 are the two promoters displaying a higher GUS expression under low Pi stress and have 3- and 4.2-fold higher expression compared to the CaMV35S promoter. In contrast to the CaMV35S promoter, the EaPHT1;2 pFL and pD2 promoter segments are susceptible to high and low Pi stress. Additionally, the promoter with a 5′ UTR region showed better expression compared to the promoter without a 5′ UTR region under the same conditions. Notably, under Pi stress and normal conditions, the pD2 promoter continued to exhibit the greatest promoter activities among all other promoters. The in silico analysis of the promoter regions suggested that the pFL promoter fragment contains three P1BS elements, eight E-Box/MYCCONSENSUSAT motifs, and one MYBST1 motif. Instead, the pD2 promoter fragment has the highest expression, containing only one E-Box and P1BS element, whereas the pD1 promoter fragment lacks the P1BS element and has no substantial expression under Pi stress. A previous study on HvPHT1 reported that the mutation of the P1BS motif resulted in a total lack of gene induction under Pi deprivation [65]. Despite the presence of fewer P1BS elements, the Pi stress response was unaffected. This implies that the gene induction is not particularly regulated by the P1BS element [66]. These findings suggest that the PIBS motif in the EaPHT1;2 promoter region is one of the significant regulators (despite its number) responsible for Pi stress, while various cis elements situated across the promoter region may also govern the Pi response. Therefore, the pFL and pD2 promoters of the *EaPHT1;2* gene may be efficient promoters for the development of transgenic expression in sugarcane and other crops under Pi stress conditions. 

## 4. Materials and Methods

### 4.1. Bacteria and Plant Materials

*Escherichia coli* strain TOP10 was used for the cloning and propagation of the plasmids, and *Agrobacterium tumefaciens* strain EHA105 was used for plant transformation. *E. coli* and *A. tumefaciens* were grown in Luria-Bertani (LB) and Yeast Extract Peptone (YEP) media at 37 °C and 28 °C with the appropriate antibiotics, respectively [67]. The tobacco seeds (*Nicotiana benthamiana* var. Bhagyalakshmi) were grown under 16 h light/8 h dark at 25 ± 1 °C and maintained for plant transformation [68]. A 6-month-old *Erianthus arundinaceus* plant was used for the study. The total genomic DNA was isolated from young leaf tissues (1 mg) of *E. arundinaceus* without midribs in liquid nitrogen using the CTAB procedure [69]. 

### 4.2. EaPHT1;2 Gene Isolation and Cloning

The *E. arundinaceus* (Ea) phosphate transporter 1;2 (PHT1;2) gene was isolated from the genomic DNA. Based on our previous study [41,42], and using the *Zea mays PHT1* genes as a reference, the gene-specific primer pair was synthesized. The polymerase chain reaction was performed using Platinum Taq DNA Polymerase (Invitrogen, Waltham, MA, USA) in a Mastercycler ^®^ nexus PCR cycler (Eppendorf, Hamburg, Germany), and the thermal cycling conditions adopted were as follows: 94 °C denaturation for 5 min, followed by 30 cycles of 94 °C for 10 s, 57 °C for 30 s, 72 °C for 1 min, and a final extension of 72 °C for 10 min. The resulting PCR product was ligated into a linearized pTZ57R/T-TA cloning vector (Thermo Fisher Scientific, Waltham, MA, USA), and transformed into *E. coli* TOP10 competent cells grown under an ampicillin selection marker. The plasmids isolated or the presence of a target gene were confirmed, and then further confirmed using the Sanger sequencing method.

### 4.3. Bioinformatics Characterization of the EaPHT1;2 Gene 

The EaPHT1;2 nucleotide sequence data were deduced using VecScreen (https://www.ncbi.nlm.nih.gov/tools/vecscreen/, (accessed on 23 November 2022)) and translated into their corresponding protein primary sequence using the Expasy translation tool (https://web.expasy.org/translate/, (accessed on 23 November 2022)) [70], and then the most suitable reading frame was selected. The consensus sequence was checked for its homology with the reported sequences presenting in the nucleotide database using the BLASTn and BLASTx programs of NCBI BLAST (https://blast.ncbi.nlm.nih.gov/Blast.cgi, (accessed on 27 November 2022)) [71]. A multiple sequence alignment tool was applied using the EBI’s ClustalW [72] and the evolutionary tree constructed by using neighbor-joining clustering in MEGA-X [73]. To compute the physicochemical properties of the *EaPHT1;2* gene, the ProtParam tool on the Expasy server was used [70]. The SMART tool (http://smart.embl-heidelberg.de/, (accessed on 28 November 2022)) [74], Pfam server (http://pfam.xfam.org/, (accessed on 2 December 2022)) [75], InterPRO (https://www.ebi.ac.uk/interpro/, (accessed on 2 December 2022)) [76], PROSITE (https://prosite.expasy.org/,(accessed on 2 December 2022)) [77], and TMHMM 2.0 tool (http://www.cbs.dtu.dk/services/TMHMM/, (accessed on 5 December 2022)) [78] were employed to determine the domains/motif and transmembrane domains of the EaPHT1;2 protein. The secondary structures were analyzed using SOPMA (https://npsa-prabi.ibcp.fr/cgi-bin/npsa_automat.pl?page=/NPSA/npsa_sopma.html, (accessed on 5 December 2022)) [79] and PsiPred (http://bioinf.cs.ucl.ac.uk/psipred/, (accessed on 5 December 2022)) [80]. The three-dimensional structures of the EaPHT1;2 proteins were predicted using the Phyre2 software V 2.0 (http://www.sbg.bio.ic.ac.uk/phyre2/, (accessed on 5 December 2022)) [81] and SWISS-MODEL (https://swissmodel.expasy.org/, (accessed on 5 December 2022)) [81]. The models obtained were subjected to PROCHECK [51] and ProSA (https://prosa.services.came.sbg.ac.at, (accessed on 5 December 2022)) [82] and were visualized using UCSF Chimera. STRING analysis (http://string-db.org/, (accessed on 6 December 2022)) [83] and subcellular localization using DeepLoc1.0. (https://services.healthtech.dtu.dk/services/DeepLoc-1.0/, (accessed on 6 December 2022)) [84] were carried out. Signal peptides, hydropathy plots, and post-translation modification site analysis were also analyzed using SignalP (http://www.cbs.dtu.dk/services/SignalP/, (accessed on 6 December 2022)) [85], ProtScale (http://web.expasy.org/protscale/, (accessed on 6 December 2022)) [70], and phosphorylation (https://services.healthtech.dtu.dk/services/NetPhos-3.1/, (accessed on 10 December 2022)) [86] and glycosylation (https://services.healthtech.dtu.dk/services/NetNGlyc-1.0/, (accessed on 10 December 2022)) [87] site prediction, respectively [41].

### 4.4. Isolation of EaPHT1;2 Promoter Region Using a Genome Walking Method

The random amplification of genome ends (RAGE) approach was used to isolate the promoter of the PHT1;2 gene from *E. arundinaceus*. The genomic DNA was isolated from the root tissues of *E. arundinaceus* using a QIAGEN DNeasy Plant Mini Kit and restricted with *DraI*, *EcoRV*, *HincII*, *PvuII*, and *SspI*, to create five genome walker libraries. Primary PCR amplifications were carried out using each of the libraries as a template and antisense gene-specific primer (GSP I) from the 5′ ends of the coding region of the *EaPHT1;2* gene sequence, as well as primary adapter primer ASP I (Appendix A), and secondary PCR amplifications were carried out using the nested adapter primer ASP II and a nested *EaPHT1;2* gene-specific primer (GPS II) [66,88,89,90]. All PCR amplifications were performed with the Taq polymerase enzyme and cloned in the pTZ57R/T-TA cloning vector, and the ligated product was mobilized into *E. coli* TOP10 [67]. Finally, the recombinants carrying the promoter region were screened and the plasmids were isolated and sequenced.

### 4.5. Bioinformatics Analysis of the Promoter Regions 

The nucleotide sequences obtained after sequencing were analyzed using NCBI BLAST analysis and the resulting EaPHT1;2 promoter sequence was further characterized using various tools and database search programs. TSS and TATA-Box were analyzed using an online neural network promoter prediction (NNPP version 2.2) (BDGP: Neural Network Promoter Prediction) (https://fruitfly.org/seq_tools/promoter.html, (accessed on 12 December 2022)) [91] and TSSPlant (http://www.softberry.com/, (accessed on 12 December 2022)) [92]. The comparative analysis was carried out using the PlantPAN 3.0 online tool (http://plantpan3.itps.ncku.edu.tw/, (accessed on 12 December 2022)) [93] and MEME tools (https://meme-suite.org/meme/tools/meme, (accessed on 12 December 2022)) (https://meme-suite.org/tools/meme, (accessed on 15 December 2022)) [94]. Using the PLACE (https://www.dna.affrc.go.jp/PLACE/, (accessed on 15 December 2022)) [95] and PlantCARE (https://bioinformatics.psb.ugent.be/webtools/plantcare/html/, (accessed on 15 December 2022)) [96] web servers, the putative *cis*-regulatory elements were determined. 

### 4.6. Vector Construction and Tobacco Transformation

The promoter region of 1.1 Kb was cloned in the plant transformation binary vector pCAMBIA1305.1 using the promoter-region-specific primers. The full-length promoter and a serious of deletions with and without the 5′ UTR regions were generated using the region-specific truncation primers (Appendix A). All the EaPHT1;2 promoter constructs were amplified using each region-specific truncation primer and replaced with the CaMV35S promoter region, which drives the GUS gene expression in the pCAMBIA1305.1 vector [89,97]. The constructs were transformed into *E. coli* TOP10 and the plasmids were confirmed using 5′ *HindIII* and 3′ *NcoI* restriction analysis and sequencing. Further, the confirmed plasmids were mobilized into *A. tumefaciens* strain EHA105 for plant transformation. Consequently, the plasmids were introduced into tobacco by following the transformation procedures as previously described [68,90].

### 4.7. Low- and High-Pi Treatments

The putative tobacco transgenic events of the independent T0 hygromycin-resistant plants expressing GUS were selected and self-fertilized to produce T1 seeds. The seeds were aseptically sown in an MS medium containing 25 mg/l hygromycin and 20-day-old-seedlings used for the Pi stress analysis [97]. The seedlings of each construct were treated with 0.1 mM (low Pi) and 1 mM KH_2_PO_4_ (high Pi) for 24 h and the samples were collected, frozen using liquid N_2_, and stored at −80 °C for further study [19,98]. 

### 4.8. Histochemical and Fluorometric β-Glucuronidase (GUS) Assays

The histochemical and fluorometric β-glucuronidase assay for GUS reporter gene expression was performed with some modifications as described in Jefferson [99]. The leaves, stems, and roots of the putative transgenic events and T1 seedlings were utilized for the GUS assays. For histochemical assay, the samples were dipped in GUS staining solution, and incubated for 12–16 h at 37 °C. After staining, sections were rinsed in 70% ethanol to remove the chlorophyll and photographed under a Leica microscope (Leica, Wetzlar, Germany). In the quantitative fluorometric GUS assays, the total protein was isolated and quantified according to the Bradford method using a bovine serum albumin (BSA) standard [100]. To quantify the GUS activity, 4-MUG (4-methylumbelliferyl b-d-glucuronide) from Sigma was used as the substrate. The fluorescence was detected using a microplate spectrofluorometer at 365 nm and 455 nm excitation and emission wavelengths, respectively. The GUS enzyme activity was presented as the pmols of 4-methylumbelliferone (MU) hydrolyzed per milligram of soluble protein per minute.

### 4.9. qRT-PCR Analysis

The total RNA was extracted from the control and stress leaves of the putative transgenic tobacco plants using a TRIzol method [101] and then treated with RNase-free DNase. The quality and quantity of the RNA were checked on 1.5% agarose gel and using a NanoDrop method (Thermo Fisher Scientific, USA). The cDNA was synthesized by utilizing a RevertAid First Strand cDNA Synthesis Kit from Thermo Fisher Scientific and according to the instructions mentioned in the protocol manual. The IDT tools were used to design the primers for the qRT-PCR (Appendix A). Three technical and biological replicates were used for the expression analysis, and the relative expression was determined by using the 2−∆∆Ct method [102]. The native control used in this study was the housekeeping gene glyceraldehyde-3-phosphate dehydrogenase (GAPDH) [103,104,105]. The qPCR reaction conditions explained by Narayan et al. [106,107,108] were followed.

## 5. Conclusions

Phosphorus (P) is indeed crucial for plant growth and development. It also regulates physiological responses and enhances abiotic stress tolerance in plants. Its deficiency can limit crop productivity, affecting both yield and quality. Phosphate transporters facilitate the uptake of phosphorus from the soil. Phosphate transporter 1 (PHT1), the widely studied P transporter, and its promoters were validated in various crop plants. But, the PHT1;2 gene promoter has been reported in very few plant species. The present study successfully isolated and functionally characterized the EaPHT1;2 gene promoter under Pi stress conditions. Full-length and truncated deletion promoters were validated using GUS reporter gene expression. The 374 bp strong and short promoter of EaPHT1;2 shows significant expression in low P conditions, and in comparison with the traditional CaMV35S promoter, EaPHT1;2 shows three-fold expression in normal conditions and two-fold expression under Pi stress conditions. Therefore, the EaPHT1;2 promoter could be exploited for the development of transgenic crops that produce good yields under P-deficient conditions. 

## Figures and Tables

**Figure 1 plants-12-03760-f001:**
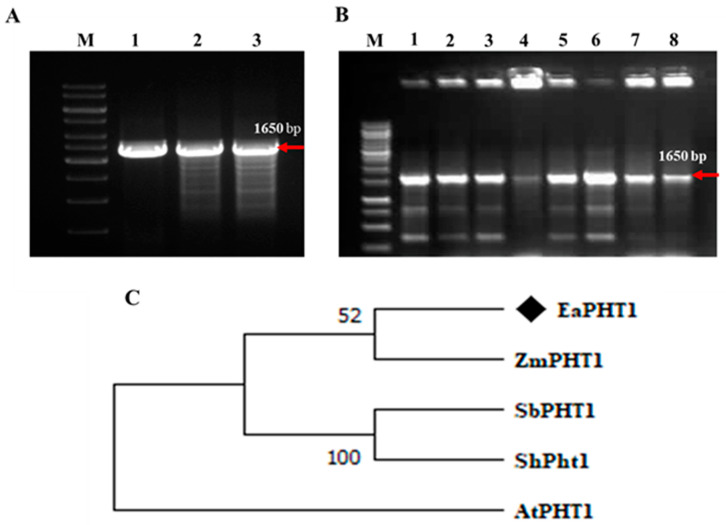
*EaPHT1;2* gene isolation and phylogenetic analysis. (**A**) *EaPHT1;2* gene amplification. Lanes: M-1 kb DNA ladder; 1 to 3, EaPHT1;2 (∼1650 bp). (**B**) *E. coli* transformants screened with *EaPHT1;2* gene-specific primer. Lanes: M-1 kb DNA ladder; 1 to 8, random transformants selected for screening showing amplification of *EaPHT1* gene. The red arrow indicates the specific amplicon of 1650 bp. (**C**) Phylogenetic analysis of *PHT1* gene obtained from *E. arundinaceus*, *Z. mays*, *S. bicolor*, *Arabidopsis*, and *Saccharum* hybrid using MEGA.

**Figure 2 plants-12-03760-f002:**
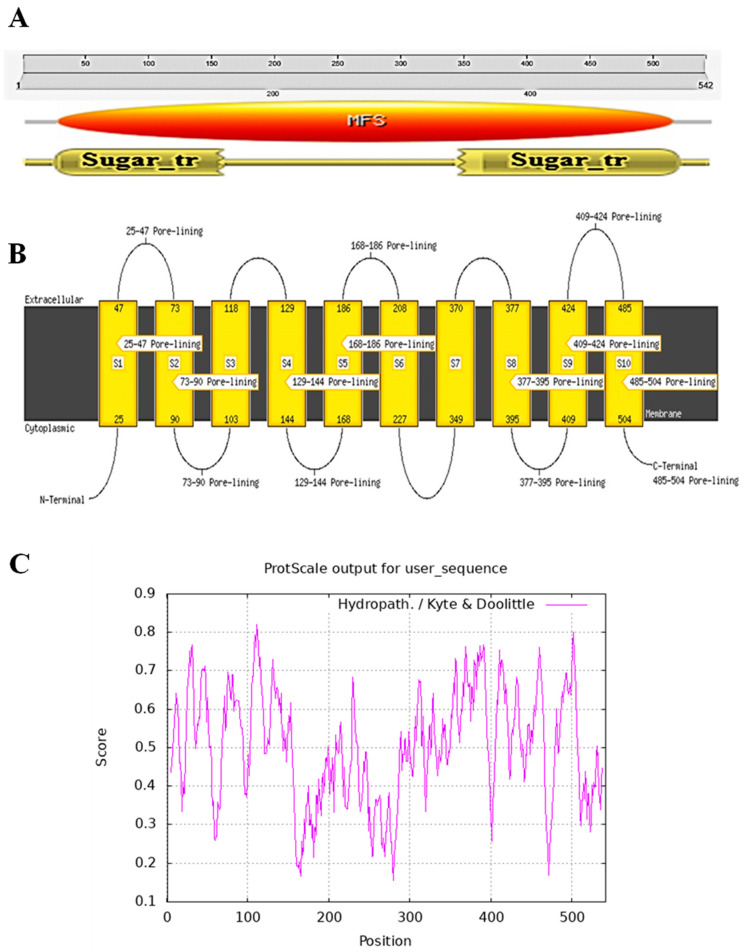
Protein domain and topology analysis of EaPHT1:2. (**A**) The major facilitator superfamily (MFS) domain, denoted in the 20-511 a.a sequence, is a gene conserved domain represented in all PHT1 genes. (**B**) The topology of EaPHT1;2 was predicted based on its protein sequence. Loops and coils are denoted by lines. The yellow blocks represent transmembrane domains S1_S10. The numbers at the top and bottom of each domain indicate the positions of amino acid residues. (**C**) Hydropathy plots of the EaPHT1;2 protein. The *X*_axis represents the location of the protein starting from the N_terminus region and the *Y*_axis indicates the hydrophobicity score of the protein. The peaks denote that EaPHT1;2 is hydrophobic in nature.

**Figure 3 plants-12-03760-f003:**
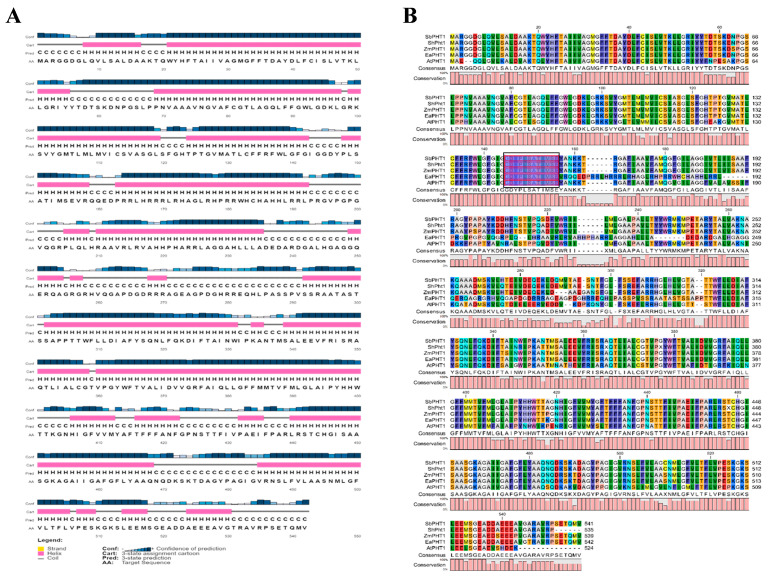
Protein structure and homologous sequence analysis. (**A**) Secondary structure of EaPHT1;2 protein predicted using PsiPred. (**B**) Homologous sequence alignment and conservation among the PHT1 proteins from *E. arundinaceus*, *Z. mays*, *S. bicolor*, *Arabidopsis*, and *Saccharum* hybrid. Violet highlighted positions in a black box represent the conserved sequences of the PHT1 gene family.

**Figure 4 plants-12-03760-f004:**
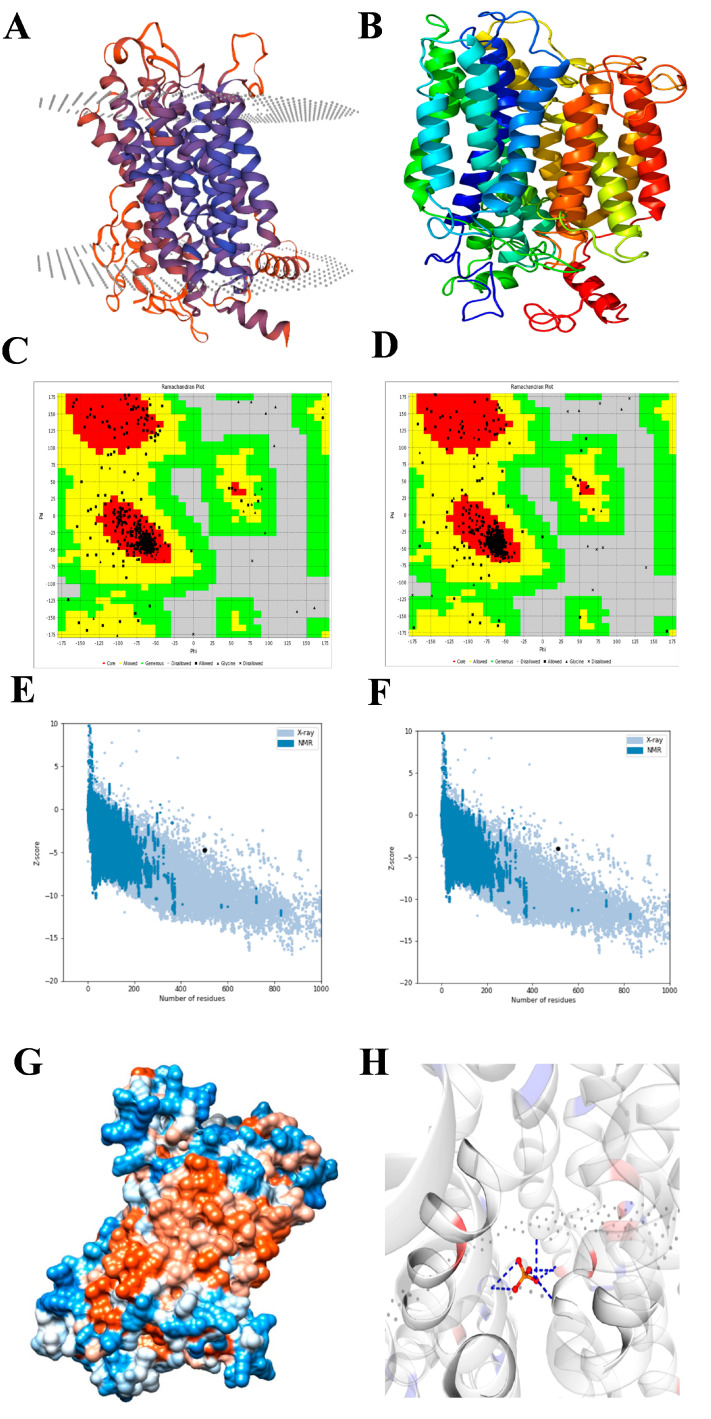
3D structural analysis and validation. (**A**) EaPHT1;2 3D structure obtained using Phyre2 analysis. (**B**) The model was obtained using SWISS_MODEL. (**C**,**D**) ramplot representation for the Phyre2 and SWISS_MODEL structure validation using VADAR. (**E**,**F**) Qualitative evaluation of EaPHT1;2 3D structure using the ProSA web server for the Phyre2 and SWISS-MODEL models. (**G**) Electrostatic surface representation of EaPTT1;2 and (**H**) the structure of EaPHT1;2 in a complex with phosphate ion (PO4) substrate.

**Figure 5 plants-12-03760-f005:**
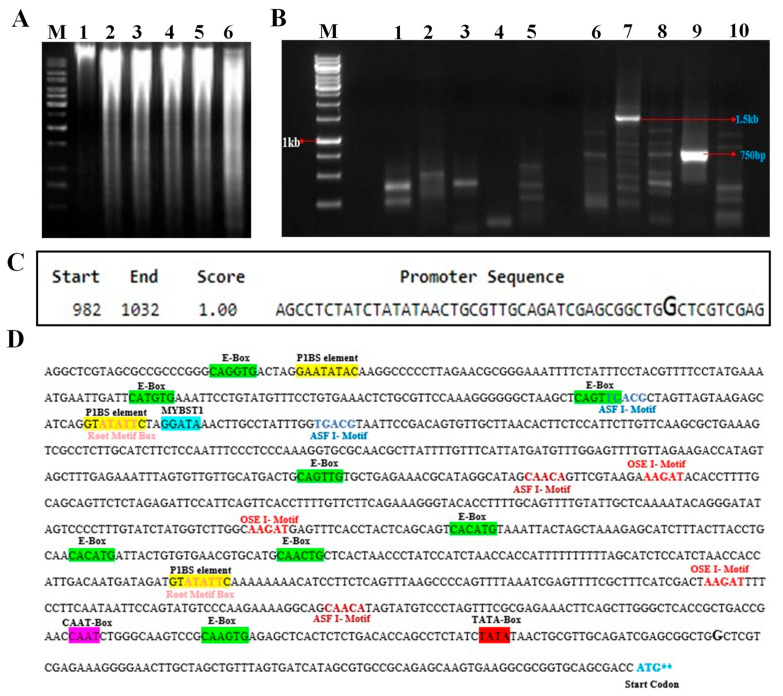
EaPHT1;2 promoter isolation and in silico validation. (**A**) Restriction of the *E. arundinaceus* genomic DNA. Lanes: M-1 kb DNA marker; 1, unrestricted gDNA; 2 to 6, restricted gDNA (lanes 2–6 as follows: DraI, EcoRV, HincII, PvuII, and SspI). (**B**) RAGE PCR amplification of EaPHT1;2 promoter; Lanes: M-1 kb DNA marker; 1–5, primary PCR products; 6–10, secondary PCR products of 1.5 kb. (**C**) Promoter sequence prediction using the NNPP tool reveals the start and end positions of the promoter together with their corresponding genomic locations and scores. The promoter region with the highest score value of 1.0 is taken, and the position of the transcription start is highlighted with a larger font size. (**D**) In silico analysis of cis elements in EaPHT1;2 promoter, common motifs, Pi-related motifs, and root-specific motifs were highlighted and marked bold in different colors.

**Figure 6 plants-12-03760-f006:**
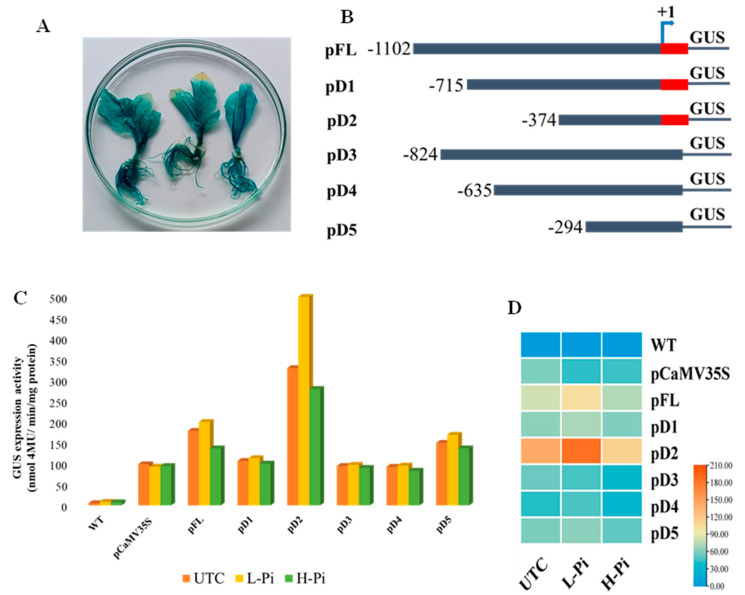
EaPHT1;2 promoters and GUS expression analysis. (**A**) Histochemical GUS expression analysis in transgenic pFL. (**B**) Systemic representation of EaPHT1;2 regulatory sequence deletion constructs. The size of each construct is given in numbers and deletions are named pD1 to pD5. The red-highlighted region denotes the 5′ UTR region. (**C**) Fluorometric analysis of putative transgenic events under Pi stress. (**D**) qRT_PCR analysis of GUS activity in transgenic tobacco lines under Pi stress. Heatmaps were created to elucidate the promoter expression levels. The color scale indicates the level of expression and the expression levels range from low expression (blue) to high expression (red). (WT_Wild_type, UTC_Untreated control, L_Pi_Low Pi, H_Pi_High Pi).

## Data Availability

The data used in this study are available in the NCBI database and as a Appendix A to this manuscript.

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
