# Peer review of "Isolation and Characterization of Erianthus arundinaceus Phosphate Transporter 1 (PHT1) Gene Promoter and 5′ Deletion Analysis of Transcriptional Regulation Regions under Phosphate Stress in Transgenic Tobacco"

_plants, 2023, doi:10.3390/plants12213760_

Round 1

Reviewer 1 Report

Comments and Suggestions for Authors

The authors isolated and characterized the potential role of Erianthus arundinaceus_EaPHT1;2 gene in conferring tolerance to phosphate stress in transgenic tobacco. Results showed that the EaPHT1;2 promoter is more efficient than the widely used CaMV35S promoter. Under phosphate stress conditions, the scientists found a significant expression of the EaPHT1;2 promoter.

The authors covered the relevant literature for inorganic phosphate intake mechanism in plants and the role of phosphate transports in its uptake, translocation, and remobilization. The role of phosphate transporters in other crops such as barley, rice, maize, millet, etc., have been reviewed in the introduction. The focus of this work on the role of PHT1 gene of sugarcane. However, little review has been conducted by the authors regarding previously work on phosphate transporters in sugarcane. This information should be developed in greater detail in the introduction section.

In the introduction, the scientific names and authors each organism should be mentioned when the organisms are first mentioned in the manuscript: E.g., sugarcace, rice, barley, maize, etc.

On the methodology, the authors should explain why 1mM KH2PO4 was considered high Pi treatment. Were there any preliminary data and work justifying this?

Figure 1 A and 1B are very difficult to read. Please increase the image resolution.

Figure 5 and 6 are missing from this manuscript.

Author Response

Here is a point-by-point response to the reviewers’ comments and concerns.

Comments from Reviewer 1

General Comment: The authors isolated and characterized the potential role of Erianthus arundinaceus_EaPHT1;2 gene in conferring tolerance to phosphate stress in transgenic tobacco. Results showed that the EaPHT1;2 promoter is more efficient than the widely used CaMV35S promoter. Under phosphate stress conditions, the scientists found a significant expression of the EaPHT1;2 promoter.

The authors covered the relevant literature for inorganic phosphate intake mechanism in plants and the role of phosphate transports in its uptake, translocation, and remobilization. The role of phosphate transporters in other crops such as barley, rice, maize, millet, etc., have been reviewed in the introduction. The focus of this work on the role of PHT1 gene of sugarcane. However, little review has been conducted by the authors regarding previously work on phosphate transporters in sugarcane. This information should be developed in greater detail in the introduction section.

Response: Thank you for your valuable comments. In sugarcane, the first comprehensive analysis of PHT genes and the comparative in-silico analysis of promoter regions was reported in our previous studies. To extend this initial work on sugarcane Phosphate transporter gene, the characterization of the promoter regions from Erianthus arundinaceus the PHT1;2 gene is reported here.

Murugan, N.; Palanisamy, V.; Channappa, M.; Ramanathan, V.; Ramaswamy, M.; Govindakurup, H.; Chinnaswamy, A. Ge-nome-Wide In Silico Identification, Structural Analysis, Promoter Analysis, and Expression Profiling of PHT Gene Family in Sugarcane Root under Salinity Stress. Sustainability 2022, 14, 15893, doi:10.3390/su142315893.

Murugan, N.; Kumar, R.; Pandey, S.K.; Dhansu, P.; Chennappa, M.; Nallusamy, S.; Govindakurup, H.; Chinnaswamy, A. In Silico Dissection of Regulatory Regions of PHT Genes from Saccharum Spp. Hybrid and Sorghum Bicolor and Expression Analysis of PHT Promoters under Osmotic Stress Conditions in Tobacco. Sustainability 2023, 15, 1048, doi:10.3390/su15021048.

Comment 1: In the introduction, the scientific names and authors each organism should be mentioned when the organisms are first mentioned in the manuscript: E.g., sugarcace, rice, barley, maize, etc.

Response: Thank you! As suggested by the reviewer, we have made the changes in the introduction section and included in the revised manuscript.

Comment 2: On the methodology, the authors should explain why 1mM KH2PO4 was considered high Pi treatment. Were there any preliminary data and work justifying this?

Response: Thank you for this suggestion. We already provided the following citations to support this statement. And in addition, we added another citation for supporting the statement.

Remy, E.; Cabrito, T.R.; Batista, R.A.; Teixeira, M.C.; Sá-Correia, I.; Duque, P. The Pht1;9 and Pht1;8 Transporters Mediate Inorganic Phosphate Acquisition by the Arabidopsis Thaliana Root during Phosphorus Starvation. New Phytologist 2012, 195, 356–371, doi:10.1111/j.1469-8137.2012.04167.x.

Chen, A.; Chen, X.; Wang, H.; Liao, D.; Gu, M.; Qu, H.; Sun, S.; Xu, G. Genome-Wide Investigation and Expression Analysis Suggest Diverse Roles and Genetic Redundancy of Pht1 Family Genes in Response to Pi Deficiency in Tomato. BMC Plant Biol 2014, 14, 61, doi:10.1186/1471-2229-14-61.

Comment 3: Figure 1 A and 1B are very difficult to read. Please increase the image resolution.

Response: Thank you! As suggested by the reviewer, we revised the Figure 1A and 1B to make them clear and included in the revised manuscript.

Comment 4: Figure 5 and 6 are missing from this manuscript.

Response: Thank you for pointing the mistake and we apologies for the inconvenience. The figure 5 and 6 have been included in the revised manuscript.

Reviewer 2 Report

Comments and Suggestions for Authors

The manuscript is well-written and the study is interesting. The authors did many in-silico examinations for EaPHT1;2 and found interesting promoter regions for potentially enhancing the expression in normal and stressful conditions. They also displayed the experiments to test the expression efficiency compared with the 35S promoter. However, the connection between the bioinformatic analyses for EaPHT1;2 and the expression regulation of EaPHT1;2 is vague. The role of EaPHT1;2 in stress response is unclear. The incorporation of the manuscript is poor as missing figures. The manuscript should be majorly revised.

1. Figures 5 and 6 are missing in the text. They also could not be found in supplementary files. Please add the figures to the revised version. As the figures are missing, the results of the experiments related to GUS, qRT-PCR and others can not be evaluated now.

2. The authors did lots of work to isolate the gene and its relative protein, subcelluar, strutrure and conservation in plants. However, the title is "Isolation and Characterization of Erianthus arundinaceus Phosphate Transporter 1 (PHT1) Gene Promoter". The authors should revise the title and also shorten the results about the characterazaiton of this gene.

3. If the expression of this gene is regulated by Pi stress, how about the protein function in Pi stress?

4. As the different length promoter regions were applied in this study, the authors should do some bioinformatic analyses for the potential cis-elements for regulating its expression.

Overall, the manuscript should be revised carefully including adding the missing figures, especially for me, the experimental parts are the most interesting and important results in this study.

Comments on the Quality of English Language

Minor revisions should be made to avoid typos.

Author Response

Here is a point-by-point response to the reviewers’ comments and concerns.

Comments from Reviewer 2

General Comment: The manuscript is well-written and the study is interesting. The authors did many in-silico examinations for EaPHT1;2 and found interesting promoter regions for potentially enhancing the expression in normal and stressful conditions. They also displayed the experiments to test the expression efficiency compared with the 35S promoter. However, the connection between the bioinformatic analyses for EaPHT1;2 and the expression regulation of EaPHT1;2 is vague. The role of EaPHT1;2 in stress response is unclear. The incorporation of the manuscript is poor as missing figures. The manuscript should be majorly revised.

Response: Thank you for your valuable comments. We have read your comments carefully and tried our best to address them one by one. We hope that the manuscript has been improved according to your suggestions and changes are included in the revised manuscript.

Major revisions

Comment 1: Figures 5 and 6 are missing in the text. They also could not be found in supplementary files. Please add the figures to the revised version. As the figures are missing, the results of the experiments related to GUS, qRT-PCR and others can not be evaluated now.

Response: Thank you for pointing the mistake and we apologies for the inconvenience caused. The figure 5 and 6 have been included in the revised version of the manuscript at the reviewers’ suggestion.

Comment 2: The authors did lots of work to isolate the gene and its relative protein, subcelluar, strutrure and conservation in plants. However, the title is "Isolation and Characterization of Erianthus arundinaceus Phosphate Transporter 1 (PHT1) Gene Promoter". The authors should revise the title and also shorten the results about the characterazaiton of this gene.

Response: Thank you! In order to support the promoter isolation and confirmation, we performed systemic characterization of the gene in this study. As a result, we didn't specify that in the title specifically. We have already condensed the characterisation of the gene into a single section with the most essential analysis. In addition, we have already shortened the points and only retained a small number in the text. While we value the reviewer's recommendation, we respectfully disagree this suggestion.

Comment 3: If the expression of this gene is regulated by Pi stress, how about the protein function in Pi stress?

Response: Thank you for the insightful query. Our research was limited to testing the PHT1 gene's promoter under Pi stress. In order to assess the PHT gene's expression pattern and perform a functional investigation of the protein under Pi stress, we did not perform any experiments. As far as I'm aware, there hasn't been any documentation of the PHT protein's role in Pi stress. However, in the future, it could be worthwhile to investigate this idea.    

Comment 4: As the different length promoter regions were applied in this study, the authors should do some bioinformatic analyses for the potential cis-elements for regulating its expression.

Response: Yes, I agree with the reviewer, this is a valuable point. In this study, the promoter's in-silico analysis was performed, and the analysis was used to determine the different length of the promoters with potential cis-elements.  As it is mentioned in figure 5, the inconvenience caused by the specific Figure's unavailability. We revised the manuscript and please refer to the revised version.

Minor revisions:

Comment 1: Minor revisions should be made to avoid typos.

Response: Thank you very much for pointing this out. We have reviewed the entire manuscript and made the revisions carefully to eliminate the typo errors.

Round 2

Reviewer 1 Report

Comments and Suggestions for Authors

The issues highlighted by the reviewers have been addressed.

Author Response

Comments from Reviewer 1

General Comment: The issues highlighted by the reviewers have been addressed.

Response: Thank you very much for your previous comments that helped us to improve this manuscript.

Reviewer 2 Report

Comments and Suggestions for Authors

I appreciate the revised version and found it is improved. There are some minor revisions that might be made.

1. Fig. 5D, please remove the "layout options" symbol inside this figure.

2. Fig. 5C, what is the purpose of the NNPP tool? From the method, the NNPP tool is used for TATA box and TSS prediction. So the legend for 5C should be clear for the prediction of TATA box and TSS in the 1.5kb promoter.

3. What are the constructs in the three tobacco plants in Fig. 6A? The authors may include the trangenic plants corresponding to 6B here if they can. Otherwise, the GUS staining plants without annotations here only show that your experiments work and do not provide information related to 6B.

4. What is the unit of the bar in Fig. 6D? Relative expression level compared to WT (normalized to 1)? The unit should be added near the bar.

Author Response

Here is a point-by-point response to the reviewers’ comments and concerns.

Comments from Reviewer 2

General Comment: I appreciate the revised version and found it is improved. There are some minor revisions that might be made.

Response: Thank you very much for your previous comments that helped us to improve this manuscript.

Minor Revision

Comment 1: Fig. 5D, please remove the "layout options" symbol inside this figure.

Response: Thank you for pointing this out. We accept and made the changes in the revised version of the manuscript. Please refer to the revised manuscript. 

Comment 2: Fig. 5C, what is the purpose of the NNPP tool? From the method, the NNPP tool is used for TATA box and TSS prediction. So the legend for 5C should be clear for the prediction of TATA box and TSS in the 1.5kb promoter.

Response: Thank you. As suggested by the reviewer, we revised this sentence as follows:

“Promoter sequence prediction by NNPP tool reveal the start and end positions of the promoter together with their corresponding genomic locations and scores. The promoter region with highest score value of 1.0 is taken and position of the transcription start is highlighted with a larger font size. ” [Pg10, Fig 5C]

Please refer to the revised manuscript. 

Comment 3: What are the constructs in the three tobacco plants in Fig. 6A? The authors may include the trangenic plants corresponding to 6B here if they can. Otherwise, the GUS staining plants without annotations here only show that your experiments work and do not provide information related to 6B.

Response: Thank you for your suggestions. Figure 6A shows the GUS staining of EaPHT1's full-length promoter (pFL) in triplicate. As a result of the full-length promoter indicating constitutive expression, additional deletion study (Figure 5B) was performed to see if the deletion series of promoter may have tissue specific expression, however we also acquired constitutive expression in the deletion analysis. If we add GUS staining pictures for deletion analysis, we observe the same type of constitutive expression GUS staining across all EaPHT1 promoters.

Comment 4: What is the unit of the bar in Fig. 6D? Relative expression level compared to WT (normalized to 1)? The unit should be added near the bar

Response: Thanks for the comment. We considered the appropriate change and made the changes in the revised version as follows:

 “qRT-PCR analysis of GUS activity in transgenic tobacco lines under Pi stress. Heatmaps were created to elucidate the promoter expression levels. The color scale indicates the level of expression and the expression levels range from low expression (blue) to high expression (red). (WT- Wild Type, UTC-Untreated Control, L-Pi – Low Pi, H-Pi – High Pi)” [Figure 6D legend].  
